# ReCo: Reminder Composition Mitigates Hallucinations in Vision-Language Models

## Abstract

Vision Language Models (VLMs) show impressive capabilities in integrating and reasoning with both visual and language data. But these models make mistakes. A common finding – similar to LLMs – is their tendency to hallucinate, i.e., generate plausible sounding text which is not grounded in the visual input, or at worst, is contradictory. A growing consensus attributes this behavior to an over-reliance on language – especially as the generation progresses, the model suffers from a "fading memory effect" with respect to the provided visual input. We study mechanisms by which this behavior can be controlled. Specifically, using ideas from geometric algebra and relational compositions, we propose the addition of a small, trainable module (named ReCo) on top of any VLM – no other modification is needed. We show that such a lightweight module is able to mitigate the fading memory effect on three of the most widely used VLMs (InstructBLIP, LlaVA, MiniGPT4), where we see performance improvements on multiple benchmarks. Additionally, we show that our module can be combined with many of the other approaches for reducing hallucination where we achieve improved results for each one.

## 1 Introduction

Given the advances in the capabilities of Large Language Models (LLMs), recent efforts have sought to extend these models to the multi-modality setting, i.e., processing and "understanding" additional modalities beyond text such as audio, images, and videos Dai et al. (2023); Liu et al. (2023); Chu et al. (2023); Fathullah et al. (2024); Zhang et al. (2023). To this end, one milestone is the development of Vision Language Models (VLMs) that can accept both images and natural language as input, and generate contextually meaningful outputs for tasks, including visual question answering and image captioning. Some prominent models are InstructBLIP Dai et al. (2023), MiniGPT4 Zhu et al. (2023); Chen et al. (2023) and Llava Liu et al. (2023; 2024b), that show strong *image+text* understanding skills. However, we know that VLMs piggyback heavily on the core capabilities of the parent LLM to which the visual representations

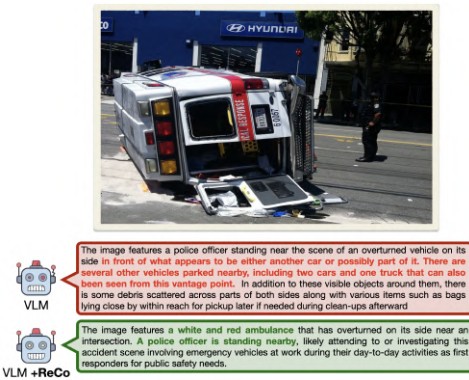

Figure 1: InstructBLIP before and after ReCo. We propose a small module that, with minimal training, it is able to effectively reduce the hallucination rate of widely used VLMs.

have been aligned. This endows sizable compute benefits – InstructBLIP Dai et al. (2023) costs about 500 GPU hours, while Llama Touvron et al. (2023) (the parent LLM), needed 180000 GPU hours. But this VLM/LLM dependence means that VLMs also inherit known weaknesses of LLMs and sometimes, these weaknesses can be magnified. One example is hallucination, i.e., generating text that is plausible but does not accurately reflect the provided input. More than a handful of results in the literature show that in many cases, a VLM appears to ignore the image and generates a description that is not influenced much at all by this extra visual input Favero et al. (2024); Leng et al. (2024); Woo et al. (2024), although this is an extreme case of hallucination. We show an example in Figure 1 where InstructBLIP Dai et al. (2023) "sees" multiple cars and trucks, although we only see an ambulance.

**Available mitigation mechanisms?** Mitigation mechanisms for the foregoing problem can be classified into two categories: **(i)** training-based methods, where the VLM is finetuned with different loss functions and/or using more suitable datasets Rafailov et al. (2023); Zhao et al. (2023); Yu et al. (2024); Jiang et al. (2024); and **(ii)** rule-based methods, where the VLM remains frozen but a new/improved generation process must be adopted. Some proposals treat the model as a black box without any access to its attention maps and its parameters Favero et al. (2024); Leng et al. (2024), whereas others use them to steer the model's generation process Huang et al. (2024); Woo et al. (2024); Liu et al. (2024c).

**Why do VLMs hallucinate?** The literature suggests that VLMs over-rely on language priors, gradually "forgetting" the visual input as text generation progresses

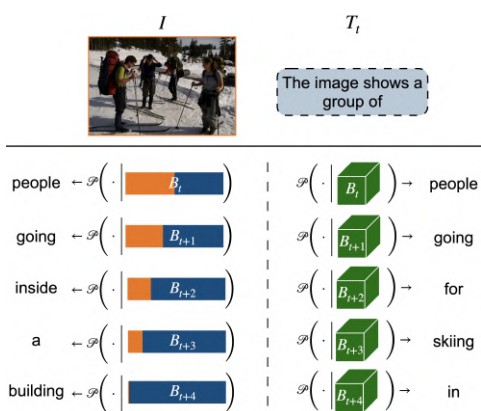

Figure 2: The "fading memory effect". **Top**: First-layer attention of the visual and textual input as the generation progresses. **Bottom**: The effect of the visual input on the logits distribution. We calculate the next token prediction with and without the visual input, and we compute the distributional difference. After the first tokens, the next token can be predicted just from the previously generated text.

Favero et al. (2024); Woo et al. (2024); Huang et al. (2024); Jiang et al. (2024); Leng et al. (2024). If next-token generation implicitly probes an internal conditional probability distribution, $\mathcal{P}(\cdot|\mathbf{\bullet})$, conditioned on both text $T_t$ and image $I$ with the right balance $\mathbf{\bullet}$, the image's role progressively diminishes to $\mathcal{P}(\cdot|\mathbf{\circ})$ and eventually to $\mathcal{P}(\cdot|\bigcirc)$, an effect Favero et al. (2024) calls the *"fading memory effect"*. We further probe this behavior: in Figure 2 (top), the total attention on visual tokens drops significantly during generation, showing that the model increasingly relies on the language prior. In Figure 2 (bottom), following Favero et al. (2024), we compute the difference in the logits distribution with and without the image. If $T_t$ is the text generated so far and $I$ is the visual input, we calculate the difference between $\mathcal{P}(y_{t+1}|T_t, I)$ and $\mathcal{P}(y_{t+1}|T_t)$, where $y_{t+1}$ denotes the next-token prediction. Using the Hellinger distance, this difference drops to almost 0 after the first 40 tokens, indicating that the image has negligible influence on subsequent tokens. Even in newer, stronger models (e.g., Qwen2.5-VL Wang et al. (2024); Qwen (2025)), the same behavior is apparent, despite all the improvements in the visual encoder, the number of image tokens, as well as the degree of fine-tuning (see Appendix A).

**The desired behavior.** A potential mitigation strategy involves a proper composition of both visual and textual embeddings, without overhauling the entire VLM architecture. If a VLM estimates the probability distribution $\mathcal{P}(y_{t+1}|B_t)$, where $B_t$ (shown as $\mathbf{\bullet}$ above) encapsulates all necessary information for the next token, then $B_t$ should combine (or **compose**) both $T_t$ and $I$, remaining sensitive to changes in *either* input for all $t$. In practice, this does not always occur, so we must intervene and carefully design $B_t$ to ensure that neither input "gets lost" (Figure 3).

**Compositionality and Geometric Algebra.** Both **(i)** compositional learning and **(ii)** geometric algebra are mature research areas Nagarajan & Grauman (2018); Aragón-González et al. (2001); Chisolm (2012) that inform our approach. *Compositionality* refers to building complex expressions from the meaning of their constituents and combination rules. It is central to visual understanding tasks such as recognizing attribute-object combinations (e.g., "spotted giraffe" Naeem et al. (2021); Mancini et al. (2021)), understanding object interactions (e.g., "person hold-

Figure 3: VLM: ideal versus practice. On the left, we show what actually happens, where at each timestep the image's influence ($I$) in orange is diminished compared to the text ($T_t$) in blue. So, the generated text is not an accurate representation of the visual input. An ideal VLM (right side) would form an object ($B_t$) that perfectly encapsulates *all* of the given input, leading to accurate generation.

ing umbrella" Krishna et al. (2017)), identifying transformations (e.g., "broken glass" Misra et al. (2017)), and parsing complex scenes Yi et al. (2018). While composition is rarely a standalone solution, it is useful for interpretability and semantic validation Chytas et al. (2024); Ganesan et al. (2021). *Geometric algebra* unifies concepts like complex numbers, quaternions, and vector algebra to express geometric relationships and transformations. Originally applied in physics, its ability to

manipulate geometric objects via operations like the geometric product (capturing inner *and* outer products) is now used in graphics Gunn & De Keninck (2019) and, more recently, machine learning Brehmer et al. (2023); Ruhe et al. (2023).

**Contributions.** This paper provides mitigation strategies for the fading memory effect by leveraging the compositional concepts above. We propose **ReCo**, a lightweight module which is data-driven but treats the VLM as a black box, combining the best of both worlds. ReCo can be easily deployed on top of any VLM during/after training and, with minimal effort, improves their hallucination behavior. We achieve promising improvements on three widely used VLMs across multiple benchmarks and datasets. Despite the small size/compute footprint, we get a performance boost without any increase in the inference time of the VLMs. Furthermore, ReCo can be combined with virtually any of the rule-based methods without any modifications.

## 2 PRELIMINARIES: GEOMETRIC ALGEBRA

Geometric (or Clifford) Algebra (GA), an extension of linear algebra, deals with a ring $\mathcal{G}$, i.e., a set of elements $\mathcal{G}$ accompanied by two operations: $\oplus$ and $\otimes$ Aragón-González et al. (2001); Chisolm (2012). The elements of GA (i.e., the elements of $\mathcal{G}$) are also vectors – $\mathcal{G}$ contains scalars (or 0-vectors), "typical" vectors (or 1-vectors) as well as 2-vectors, 3-vectors, and so on. The subset of $\mathcal{G}$ that contains all $k$-vectors is usually denoted as $\mathcal{G}_k$.

**Multi-vectors.** A $k$-vector can be defined as the geometric product of $k$ orthogonal 1-vectors, i.e., if $\{\mathbf{v}_1, \cdots, \mathbf{v}_k\} \in \mathcal{G}_1$ and $\mathbf{v}_i \perp \mathbf{v}_j, \ \forall i \neq j \in [k] \times [k]$, then

$$\mathbf{v}_1 \otimes \cdots \otimes \mathbf{v}_k = \mathbf{v}^{(k)} \in \mathcal{G}_k \tag{1}$$

Alternatively, each $k$-vector can be defined as the "wedge" ($\wedge$) product of $k$ 1-vectors, where the wedge product is:

$$\mathbf{v}_1 \wedge \cdots \wedge \mathbf{v}_k = \frac{1}{k!} \bigoplus_\sigma \text{sign}(\sigma) \otimes \mathbf{v}_{\sigma(1)} \otimes \cdots \otimes \mathbf{v}_{\sigma(k)}, \tag{2}$$

where $\sigma$ denotes a permutation from the symmetric group. The sign is $+1$ (or $-1$) for even (or odd) permutations and the following equivalence holds:

$$\mathbf{v}_1 \otimes \cdots \otimes \mathbf{v}_k = \mathbf{v}_1 \wedge \cdots \wedge \mathbf{v}_k \Leftrightarrow \mathbf{v}_1 \perp \cdots \perp \mathbf{v}_k \tag{3}$$

**Geometric operations.** A 2-vector represents a plane, a 3-vector a 3D object, and so on—unlike the inner product, which reduces vectors to a scalar with no recovery of inputs. The geometric product resembles and generalizes the outer product. These GA operations and multi-vectors motivate our instantiation: a systematic method to *fuse* or *compose* vectors beyond the common weighted averaging used in most architectures – this is the relevance to our "reminder composition". Wattenberg & Viégas (2024); Vaswani et al. (2017); Gholamalinezhad & Khosravi (2020).

**Practical implementations and limitations.** While GA gives us the necessary axioms/properties, it does not provide a specific, practical instantiation. In general, a correct (but efficient) implementation in low dimensions ($\leq 6$) requires intensive effort, see De Keninck (2024). In general, since we deal with vectors, the typical choice would be the tensor product, which is infeasible for high dimensional vectors or multi-products (exponential memory), see Gunn & De Keninck (2019), where quaternions are used for a computer graphics use case. We shift our focus to ideas that can scale while preserving, to the extent possible, the general framework of GA.

**Vector Symbolic Architectures.** Wattenberg & Viégas (2024) recently introduced ways to combine vectors beyond linear combinations. Earlier works Plate (1995); Gosmann & Eliasmith (2019) formalized this under *Vector Symbolic Architectures* (VSA), named for their use of high-dimensional vectors and operations ($\otimes$, $\oplus$) resembling bind and bundle in symbolic and connectionist AI Thomason (2009); Fodor & Pylyshyn (1988); Smolensky (1990); Touretzky & Hinton (1985). For a detailed overview, see Schlegel et al. (2022). Recent works Wolff et al. (2018) describe how these these ideas are linked to physics. This allows us to adopt these operations directly for our application.

**Relevance of these concepts.** Our main hypothesis is that the *fading memory effect* can be mitigated through a *composition* of the textual and visual information. The logical question then is how this composition can be designed, say, first conceptually and then for practical use. Theoretically, GA

provides us a structured framework for composition problems, enabling the integration of modalities such as image and text, **without** the need of ad-hoc solutions. Practically, VSA offers efficient implementations with favorable computational complexity, memory usage, and noise tolerance while preserving GA's properties.

## 3  REMINDER COMPOSITION (ReCo)

**Overview.** We mitigate the *fading memory effect* by computing $B_t$ as an explicit composition of $T_t$ and $I$ (Figure 3). If $B_t$ is designed as an (almost) lossless multi-vector of the generated text and input image, performance improves. Our approach explicitly forms this multi-vector of text and image information *before* decoding to the token space.

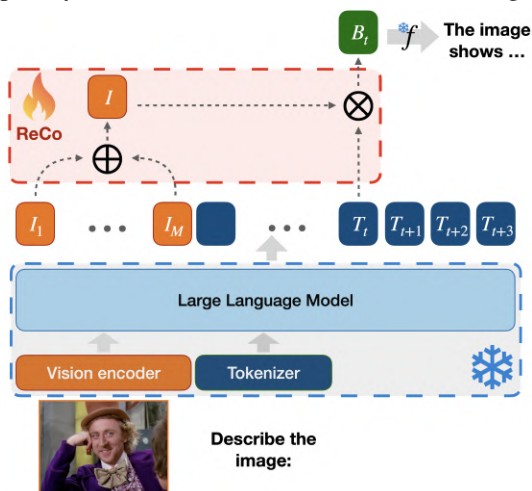

Let $[T_t]_{t=1}^N$ be the VLM's predicted embeddings for each one of the $N$ generated tokens which are then fed into the LLM head ($f$) to form the next token $[w_t]_{t=1}^N$. $T_t$ corresponds to the hidden state of the LLM at the step $t$ (usually denoted as $\mathbf{h}_t$ but we use $T$ here to mnemonically suggest "text"). Additionally, let $[I_j]_{j=1}^M$ be the $M$ output embeddings that correspond to the $M$ image embeddings that are fed to the VLM. An ideal VLM should closely capture the composition:

$$T_t = g\left( T_{t-1} \otimes \left( \bigoplus_{j=1}^M I_j \right) \otimes \mathbf{p}_t \right) \qquad (4)$$

which states that the next token depends on the previous one *composed* with all of the provided image information, with $g$ denoting a learnable transformation of the multi-vector and $\mathbf{p}_i$ denoting some extra optional information about the index of the current token. The next token corresponds to $w_t = f(T_t)$ but now the image effect, by design, **cannot** be neglected. Notice that (4) simply re-formulates what we have already observed about $p(y_{t+1}|T_t, I)$ and $p(y_{t+1}|T_t)$ in Figure 3, showing how $B_t$ should actually behave in theory:

Figure 4: **ReCo overview**. The VLM is treated as a black box, modifying the next-token embedding by combining the multi-vector of visual tokens and the current token prediction. First, we bundle the image tokens $[I_j]_{j=1}^m$ into a single vector $I$, then bind it with $T_t$ to form $B_t$, ensuring the image's influence. Finally, the frozen prediction head $f$ outputs the next token $w_t$.

$$B_t = g\left( T_{t-1} \otimes \left( \bigoplus_{j=1}^M I_j \right) \otimes \mathbf{p}_t \right) \rightarrow w_t = f(B_t) \ \propto \mathcal{P}(y_{t+1}|B_t) \qquad (5)$$

Therefore, we propose to explicitly modify the VLM's output so that it corresponds to the composition as described in (4). Based on Wattenberg & Viégas (2024), we define the geometric product as the Matrix Binder operation, allowing us to mitigate the fading memory problem by adding only a small trainable layer on top of a frozen, black-box VLM as:

$$B_t = W_T T_t + W_I \left( \bigoplus_{j=1}^M I_j \right) \qquad (6)$$

The modifications in any VLM's codebase consist of the addition of **only two extra lines of code** without any other changes to the model's "internals" (see Figure 5).

Many composition rules have been proposed in the literature but our choice above was driven by two reasons:

**(a)** The matrices $W_T$, $W_i$ can be trained alongside the VLM, allowing our modification to be integrated into any model without altering the training process while, at the same time, leveraging the explicit composition rule during inference and generation.

Figure 6: POPE Li et al. (2023) results on MiniGPT4 Zhu et al. (2023). The unmodified version is often unable to comprehend the question and outputs an unrelated answer that does no contain either "Yes" or "No". On the contrary, ReCo provides the model the ability to answer such questions.

**(b)** Our modified VLM extends the original model. Setting $W_T = I$ and $W_I = 0$ restores the original VLM, making the modified solution space a *strict superset*. Thus, using ReCo in training retains all original capabilities while potentially improving hallucination mitigation.

We should note that our formulation depends on a specified composition strategy. We used one which is mathematically sound but also efficient, but the operations can be upgraded.

**VLM is a black-box.** Notice that (6) and the corresponding code (Figure 5) involves only the output layer of the VLM. This means that, in practice, *no access to the model* is needed. The output embeddings of the model can be calculated offline, and then we train only the few extra parameters of ReCo in a matter of minutes on any commodity GPU, without even needing to load the entire VLM in memory and performing multiple inference passes through it. This allows us to benefit from additional training data and more suitable training tactics (similar to fine-tuning approaches, e.g., Zhao et al. (2023)) while treating the VLM

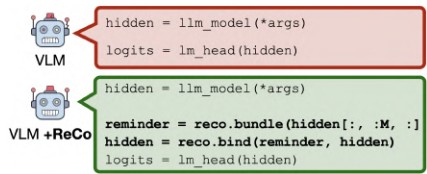

Figure 5: ReCo in practice. No access to the LLM is required and the only change is the modification of the prediction head with the addition of a "preprocessing" step.

as a frozen black box (akin to decoding strategies, e.g., Woo et al. (2024); Favero et al. (2024)). A comparison with existing solutions is in the appendix.

## 4 EXPERIMENTS

Before diving into the technical details and comprehensive analysis,, we highlight some of our main findings which we will analyze in detail shortly. These results offer a high-level overview of the most significant outcomes of our work, which will be examined and discussed in depth below.

**(a)** ReCo leads to noticeable improvements across **all** of the VLMs we evaluated. In every case, it effectively reduces the rate of hallucinations, demonstrating its robustness and general applicability.

**(b)** ReCo works seamlessly with other methods for reducing hallucinations, making it easy to integrate into different systems. Further, when combined with other approaches, it leads to even better performance, showing that the benefits of ReCo and other techniques can complement each other.

**(c)** ReCo enables the model to effectively "recover" from an initial hallucination, distinguishing it from baseline VLMs which often persist in elaborating on hallucinated content. This ability to self-correct contributes to more coherent and accurate model outputs.

**(d)** Our quantitative and qualitative analyses show that ReCo goes beyond just fixing object-related hallucinations. It also helps correct mistakes related to the overall structure of the image and the specific attributes or features of the objects themselves.

### 4.1 EXPERIMENTAL SETUP

**Datasets and Training.** We use the same training procedure across all experiments, training ReCo with HA-DPO Zhao et al. (2023) while keeping the rest of the model frozen, unlike native HA-DPO, which finetunes the entire VLM. The dataset consists of quadruples $(I, P, C, R)$, where $I$ is the

Table 1: CHAIR Rohrbach et al. (2018) results for both InstructBLIP Dai et al. (2023) and MiniGPT4 Zhu et al. (2023). *Original* stands for the unmodified VLM, while VCD Leng et al. (2024), M3ID Favero et al. (2024), and AvisC Woo et al. (2024) are the three baselines we consider. In all four models, the addition of ReCo improves significantly the performance as the generation progresses, reducing $CHAIR_s$ as much as 44% and $CHAIR_i$ 30%, with 32, 64, 128, 256, 512, and 1024 standing for the maximum allowed number of generated tokens.

| Model | | | $CHAIR_s(\downarrow)$ | | | | | | $CHAIR_i(\downarrow)$ | | | | | | Avg length |
|---|---|---|---|---|---|---|---|---|---|---|---|---|---|---|---|
| | | 32 | 64 | 128 | 256 | 512 | 1024 | 32 | 64 | 128 | 256 | 512 | 1024 | |
| InstructBLIP | Original | 8.6 | 22.0 | 37.4 | 37.2 | 37.2 | 37.2 | 4.8 | 8.3 | 11.8 | 12.2 | 12.2 | 12.2 | 456 |
| | **+ReCo** | +0.8 | +0.2 | -7.0 | -8.8 | -8.8 | -8.8 | +0.5 | +0.3 | -1.1 | -1.6 | -1.6 | -1.6 | 350 |
| | VCD | 9.6 | 19.2 | 37.6 | 36.0 | 36.0 | 36.0 | 5.0 | 6.7 | 11.2 | 11.4 | 11.4 | 11.4 | 427 |
| | **+ReCo** | -1.0 | -0.8 | -8.6 | -8.6 | -8.6 | -8.6 | -0.5 | +0.5 | -1.5 | -2.8 | -2.8 | -2.8 | 329 |
| | M3ID | 8.6 | 22.2 | 31.2 | 32.8 | 32.8 | 32.8 | 3.9 | 7.1 | 9.7 | 9.6 | 9.6 | 9.6 | 410 |
| | **+ReCo** | -2.0 | -2.6 | -5.8 | -7.4 | -7.4 | -7.4 | -0.8 | -0.7 | -1.3 | -1.2 | -1.2 | -1.2 | 294 |
| | AvisC | 8.2 | 17.6 | 32.2 | 33.0 | 33.0 | 33.0 | 4.2 | 6.4 | 9.3 | 9.1 | 9.1 | 9.1 | 408 |
| | **+ReCo** | -2.2 | +0.2 | -9.2 | -8.8 | -8.8 | -8.8 | -1.5 | -0.4 | -2.2 | -1.4 | -1.4 | -1.4 | 311 |
| MiniGPT4 | Original | 9.6 | 19.4 | 31.4 | 36.2 | 37.6 | 37.6 | 4.9 | 7.5 | 9.9 | 11.7 | 12.8 | 13.4 | 374 |
| | **+ReCo** | +2.6 | +2.2 | -2.8 | -4.6 | -5.4 | -4.8 | +1.1 | +1.3 | -0.4 | -1.5 | -2.6 | -3.2 | 307 |
| | VCD | 10.0 | 20.8 | 31.0 | 35.8 | 36.6 | 36.6 | 5.1 | 8.5 | 10.5 | 12.1 | 13.0 | 13.5 | 336 |
| | **+ReCo** | +0.8 | -2.2 | -7.6 | -9.6 | -8.8 | -8.4 | +0.1 | -0.4 | -0.7 | -1.2 | -1.5 | -1.9 | 249 |
| | M3ID | 9.4 | 22.2 | 39.2 | 48.5 | 49.5 | 49.5 | 4.6 | 9.1 | 12.3 | 15.5 | 15.9 | 16.0 | 372 |
| | **+ReCo** | +2.6 | +3.2 | -2.2 | -9.1 | -9.9 | -9.9 | +1.0 | +0.6 | +0.2 | -1.4 | -1.6 | -1.7 | 380 |
| | AvisC | 10.8 | 20.8 | 30.8 | 41.4 | 46.4 | 48.2 | 5.2 | 7.9 | 10.1 | 12.1 | 14.6 | 16.0 | 277 |
| | **+ReCo** | -1.8 | +4.4 | -8.6 | -17.2 | -20.8 | -22.2 | -0.4 | -1.1 | -2.1 | -3.1 | -4.5 | -5.1 | 220 |

image, *P* is its associated prompt or question, and *C* and *R* are the *Chosen* and *Rejected* answers for the Direct Preference Optimization Rafailov et al. (2023). The images are a small subset of Visual Genome Krishna et al. (2017) (only 1853 in total), with *C* and *R* generated using GPT-4 through a three-stage process Zhao et al. (2023).

**Evaluation.** To systematically evaluate ReCo, we use **five** benchmarks: CHAIR Rohrbach et al. (2018), POPE Li et al. (2023), AMBER Wang et al. (2023), HallusionBench Guan et al. (2024), and MME Fu et al. (2023).**(a)** POPE asks binary existence questions like *"Is there a ⟨object⟩ in the image?"*. **(b)** CHAIR measures hallucination rates in image captioning. **(c)** AMBER assesses both discriminative and generative capabilities with binary and open-ended questions such as *"Is the umbrella open?"* and *"Describe the image."*. **(d)** HallusionBench and MME test various discriminative questions, evaluating accuracy and accuracy+ (see appendix). CHAIR Rohrbach et al. (2018) and AMBER Wang et al. (2023) use MSCoco Lin et al. (2014), while POPE Li et al. (2023) is based on MSCoco, A-OKVQA Schwenk et al. (2022), and GQA Hudson & Manning (2019).

**Baselines.** Beyond comparing with unmodified VLMs (InstructBLIP Dai et al. (2023), MiniGPT4 Zhu et al. (2023), and LlaVA Liu et al. (2023)), we evaluate three recent hallucination mitigation methods: M3ID Favero et al. (2024), VCD Leng et al. (2024), and AvisC Woo et al. (2024). We report their performance before and after integration with ReCo to assess complementarity. Due to space constraints and the similar performance of InstructBLIP and LlaVA, detailed LlaVA results are provided in the appendix.

## 4.2 RESULTS

Our extensive experiments show that ReCo can be deployed easily with any VLM and works well across multiple benchmarks. Additionally, it can be combined efficiently with other methods to strengthen the results further. Below, we analyze the results on each benchmark separately.

### 4.2.1 CHAIR: EXPERIMENTAL EVALUATIONS

We prompt the model with "Describe the image.", without providing any additional information about the length of the description (e.g., "Provide a short description of the image"). In table 1 (and Tab. 4 in the appendix), we present the improvement we achieve for both $CHAIR_s$ and $CHAIR_i$ metrics as we increase the length of the generated response.

ReCo helps – reducing CHAIRs by up to 44% and CHAIRi by 30%. While baselines sometimes perform better with fewer tokens, this is often due to extra characters before the actual text, affecting the effective output length. The key trend is that as token count increases, ReCo consistently improves both metrics *significantly*.

Table 2: POPE Li et al. (2023) results for MSCoco Lin et al. (2014) and A-OKVQA Schwenk et al. (2022). In all cases, ReCo provides a significant performance boost to all the methods (where *Original* stands for the unmodified VLM).

| Model | | MSCoco Lin et al. (2014) | | | | | | A-OKVQA Schwenk et al. (2022) | | | | | |
|---|---|---|---|---|---|---|---|---|---|---|---|---|---|
| | | Random | | Popular | | Adversarial | | Random | | Popular | | Adversarial | |
| | | Acc (↑) | F1 (↑) | Acc (↑) | F1 (↑) | Acc (↑) | F1 (↑) | Acc (↑) | F1 (↑) | Acc (↑) | F1 (↑) | Acc (↑) | F1 (↑) |
| InstructBLIP | Original | 82.8% | 82.8% | 77.6% | 79.3% | 74.6% | 76.8% | 80.6% | 82.0% | 73.9% | 77.4% | 67.8% | 73.3% |
| | +ReCo | +1.3% | +0.3% | +1.5% | -0.6% | +2.8% | +0.5% | +3.4% | +0.5% | +6.9% | +2.2% | +8.1% | +2.1% |
| | VCD | 83.2% | 83.2% | 77.6% | 78.9% | 75.8% | 77.5% | 82.0% | 83.2% | 74.9% | 77.9% | 69.7% | 74.6% |
| | +ReCo | +1.6% | +0.2% | +2.9% | +0.8% | +3.6% | +1.3% | +3.2% | +0.3% | +8.0% | +3.7% | +7.1% | +1.7% |
| | M3ID | 84.1% | 84.2% | 77.9% | 79.3% | 75.0% | 77.2% | 82.7% | 83.8% | 75.9% | 78.7% | 67.9% | 73.6% |
| | +ReCo | +1.1% | -0.3% | +3.0% | +0.9% | +4.9% | +1.3% | +3.1% | +0.4% | +6.8% | +2.5% | +9.2% | +3.1% |
| | AvisC | 88.3% | 87.8% | 81.9% | 82.3% | 79.5% | 80.3% | 86.2% | 86.7% | 80.4% | 82.1% | 71.5% | 75.8% |
| | +ReCo | -0.9% | -1.2% | +1.7% | +0.7% | +2.2% | +1.2% | +1.1% | -0.6% | +3.0% | +0.5% | +5.5% | +1.6% |
| MiniGPT4 | Original | 55.9% | 36.1% | 53.3% | 34.8% | 53.4% | 34.8% | 55.3% | 36.6% | 55.7% | 33.0% | 55.3% | 32.8% |
| | +ReCo | +2.0% | +13.9% | +3.1% | +14.3% | +2.2% | +9.8% | +6.8% | +7.4% | +6.1% | +10.8% | +3.7% | +9.2% |
| | VCD | 58.3% | 44.1% | 55.6% | 42.5% | 55.3% | 42.4% | 60.9% | 48.9% | 57.2% | 46.6% | 56.6% | 47.0% |
| | +ReCo | +0.9% | +5.7% | +2.0% | +6.3% | +1.7% | +6.1% | +0.1% | -2.0% | +2.6% | -0.3% | +1.3% | -1.1% |
| | M3ID | 57.6% | 41.7% | 55.0% | 40.3% | 54.3% | 39.9% | 59.5% | 38.9% | 56.1% | 37.0% | 56.0% | 36.9% |
| | +ReCo | +0.1% | +7.5% | +1.4% | +8.2% | +1.3% | +8.1% | +1.7% | +2.4% | +4.9% | +4.2% | +2.4% | +2.7% |
| | AvisC | 64.6% | 62.9 | 61.7% | 61.0% | 59.2% | 59.5% | 61.5% | 50.3% | 65.3% | 56.3% | 60.3% | 52.3% |
| | +ReCo | -2.1% | -7.1% | +1.4% | -5.9% | +1.1% | -3.8% | +0.6% | +0.7% | -0.9% | -2.8% | +0.4% | -2.0% |

**Analysis.** Like other mitigation methods, ReCo cannot fully eliminate hallucinations. However, its impact is evident in the significant improvement on CHAIR$_s$. While hallucinations may still occur, ReCo prevents the model from fixating and building upon them. For example, in Figure 1, a model might mistakenly generate *intersection*, but unlike the unmodified VLM, it does not continue elaborating on this error. This is due to ReCo's image "reminder", which helps steer generation back toward accuracy, counteracting the over-reliance on language priors observed in VLMs (fig. 2).

### 4.2.2 POPE: EXPERIMENTAL EVALUATIONS

Since POPE Li et al. (2023) relies on binary (Yes/No) questions about objects in images, the *fading memory effect* is expected to be less severe. However, it is important to assess ReCo's performance here and determine if it enhances the results. Table 2 presents Accuracy and F1 scores across three question types (Random, Popular, and Adversarial) for MSCoco Lin et al. (2014) and A-OKVQA Schwenk et al. (2022). Additional results for LlaVA Liu et al. (2023) and GQA Hudson & Manning (2019) are available in the appendix.

**Analysis.** We can observe that ReCo consistently improves the performance across all models, questions, and datasets and, more importantly, it does not overfit the CHAIR benchmark, which is more related to the long-range dependency between the input image and the generated text. For MiniGPT4 specifically, ReCo is **one of the only few** black-box approaches that allows the model to comprehend and answer existence questions correctly, as shown in fig. 6 and further explained and analyzed in section 4.2.5.

### 4.2.3 AMBER: EXPERIMENTAL EVALUATIONS

AMBER Wang et al. (2023) evaluates both the generative and the discriminative capabilities of a VLM, by asking open-ended questions (e.g., "Describe the image") as well as multiple yes/no questions (e.g., "Is the sky sunny?", or "Is there a direct contact between the car and the tree?").

**Analysis.** ReCo provides a significant performance boost for **all** baselines, demonstrating that, despite the minimal training and modifications we need, it is a valuable add-on for hallucination mitigation in both task families.

### 4.2.4 HALLUSIONBENCH: EXPERIMENTAL EVALUATIONS

HallusionBench Guan et al. (2024) evaluates the discriminative capabilities of a VLM by providing it with (modified) images of charts, tables, and maps and asking related questions (see appendix for examples).

Table 3: AMBER Wang et al. (2023) (left) and HallusionBench Guan et al. (2024) (right) results for InstructBLIP Dai et al. (2023), MiniGPT4 Zhu et al. (2023), and Llava Liu et al. (2023). ReCo consistently improves the performance of all models.

| | AMBER (↑) | | | HallusionBench (↑) | | |
|---|---|---|---|---|---|---|
| Model(+ReCo) | InstructBLIP | MiniGPT4 | LLaVA | InstructBLIP | MiniGPT4 | LLaVA |
| Original | 81.4 (+3.1) | 51.5 (+32.8) | 78.2 (+2.3) | 50.5% (+3.6%) | 46.0% (+3.8%) | 51.9% (+6.9%) |
| VCD | 82.6 (+3.8) | 58.3 (+23.7) | 77.9 (+2.1) | 49.8% (+1.0%) | 44.7% (+6.1%) | 49.7% (+8.4%) |
| M3ID | 83.0 (+3.5) | 49.0 (+33.1) | 77.9 (+2.9) | 49.6% (+1.7%) | 46.3% (+4.9%) | 53.4% (+7.0%) |
| AvisC | 85.0 (+2.2) | 66.0 (-02.6) | 79.3 (+2.1) | 47.6% (+3.8%) | 44.0% (+5.4%) | 51.5% (+6.6%) |

**Analysis.** Table 3 shows the average accuracy on the HallusionBench. ReCo *always* improves the results, irrespective of which underlying VLM and baseline is used. More importantly, in many cases the rest of the baselines fail to improve the unmodified VLM (e.g., all baselines perform worse in InstructBLIP). This underscores the fact that ReCo is more robust across different and diverse questions, like the ones in HallusionBench.

### 4.2.5 ON THE DISCRIMINATIVE CAPABILITIES OF MINIGPT4

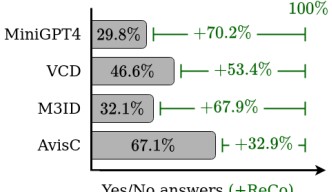

Figure 7: MiniGPT4 out of the box as well as other methods are not able to answer properly (with a *Yes* or *No*) discriminative questions in most cases. In contrast, ReCo offers the model the ability to comprehend/answer such questions.

As we see in Figure 6, MiniGPT4 is not able to answer existence questions in most of the cases. During AMBER evaluations, we observed that this deficiency extends to all types of discriminative questions. In Figure 7 we show that MiniGPT4, as well as its modifications, do not possess the ability to answer many discriminative (i.e., yes/no) questions, and a failure example is presented in Figure 8. As shown, MiniGPT4 is able to understand and answer with either "Yes" or "No" (independent of the true response label) in less than 30% of the cases, and, while the percentage increases, existing works also face the same difficulty. This, however, means that the results obtained by all baselines on POPE and AMBER do not accurately reflect reality, as these models are not well suited for such questions and small modifications to the evaluation scripts can lead to very different values of accuracy and F1 score. On the contrary, ReCo is able to answer such questions with a 100% rate in all cases, offering a useful new capability to the underlying VLM.

### 4.2.6 STRUCTURAL HALLUCINATIONS

CHAIR Rohrbach et al. (2018) and POPE Li et al. (2023) evaluate only object-related hallucinations. AMBER Wang et al. (2023) evaluates the model only in the context of yes/no questions although some of its questions are about object relationships and their attributes. However, in our experiments, we observed that beyond a significant reduction in such hallucinations, ReCo is also able to effectively reduce structure-related hallucinations (e.g., relative positions of the objects, object attributes, and so on) in the images' description. In Figure 9, we show a few examples where the effect of ReCo is apparent in reducing such hallucinations also. From relative positions to textures and text signs, the improvement of ReCo is apparent

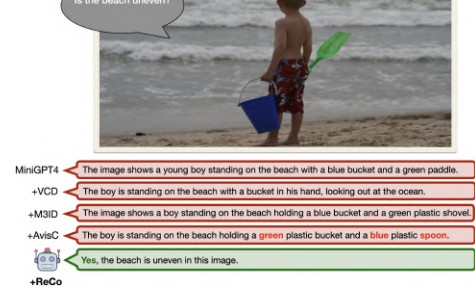

Figure 8: Failure of all MiniGPT4-based models but ReCo to answer the AMBER Wang et al. (2023) questions coherently. All models describe the image (with some of them getting the details wrong) although the question is a binary (yes/no) one (whose true label is "Yes").

in all these cases. Interestingly, we can observe that ReCo does not change the output of the VLM completely, rather it "intervenes" only when actually needed, leaving the remainder of the output almost intact. This is of course a by-product of the fact that we treat the VLM as a frozen black-box and we only change the input to the prediction head.

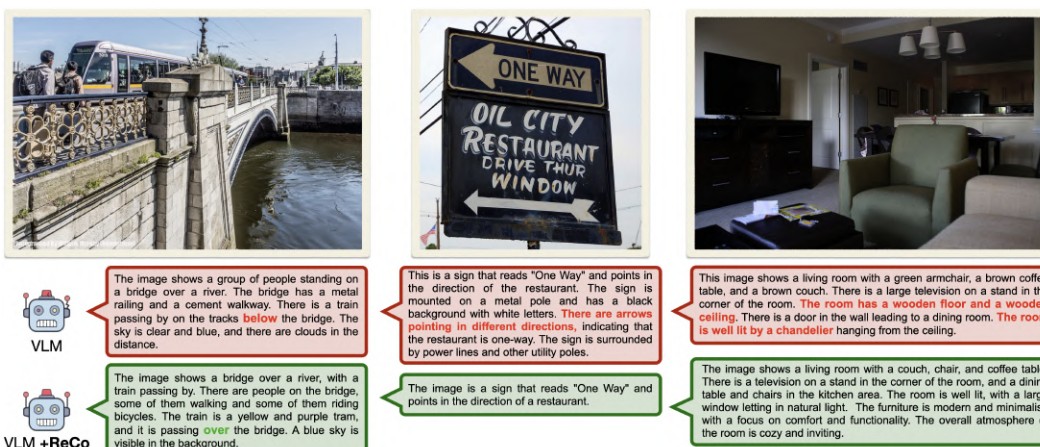

Figure 9: Structural hallucinations: MiniGPT4 Zhu et al. (2023) before *(red)* and after ReCo *(green)*. The unmodified VLM tends to get small details about the scene wrong, like the texture of the floor or the written signs that are depicted in the image. Enabling ReCo fixes such mistakes.

## 5 RELATED WORK

**Hallucinations mitigation.** Despite their unique capabilities, it is well-known that VLMs hallucinate during the generation process. Multiple works have proposed modified loss functions, optimization schemes, and new datasets Rafailov et al. (2023); Zhao et al. (2023); Jiang et al. (2024); Yu et al. (2024) that can improve the performance of the models, albeit extensive re-training of the whole (or a large part) architecture is sometimes needed. A different line of work has proposed modified generation processes: usually, a contrastive decoding approach that "boosts" the influence of the visual input to the output Huang et al. (2024); Favero et al. (2024); Woo et al. (2024); Leng et al. (2024). A detailed review can be found in Liu et al. (2024a). Like many of these works, our work also treats the model as a frozen, black box and it intervenes only in the next token prediction. However, similar to the first family of approaches, it is data-driven and it can benefit from the newly proposed datasets and optimization techniques.

**Vector Symbolic Architectures.** Ideas describing Vector Symbolic Architectures (VSAs) date back to the 1990s where one of the focus was on introducing operations for efficiently combining (binding) multiple vectors together Plate (1995); Gosmann & Eliasmith (2019). The motivation stems from ideas in Symbolic AI where one sought to combine symbols into more complicated sentences but VSAs take also advantage of the representation power of high-dimensional vectors. Multiple works proposed different instantiations of the bind and bundle operations, each one with different performance profiles. A complete review can be found in Schlegel et al. (2022). Recently, some results have infused such ideas into deep learning Wolff et al. (2018), achieving a more explicit compositional behavior, in problems ranging from extreme multi-label classification Ganesan et al. (2021) to a reformulation of the attention mechanism Alam et al. (2023).

## 6 CONCLUSIONS

While VLMs should operate in a way that the next token generation is conditioned on an entity that is a composition of the visual and textual input, this is often not the case in practice. Our work describes Reminder Composition (ReCo), a modification to the output of any VLM, which explicitly composes the visual and textual information. This modification requires minimal training, and despite treating the VLM as a black box is able to significantly improve VLM's forgetting behavior. As a result, we can significantly reduce the hallucination rate of these models. Additionally, ReCo is compatible with other works in hallucination mitigation, and their combination further improves the results across all models and benchmarks.

**Impact & Limitations.** ReCo can greatly improve VLMs and help in their smooth and non-harmful usage by all types of users. However, like other mitigation methods, it is not a silver bullet and can benefit from richer compositional rules, access to the model's hidden states, and a deeper integration with the VLM pretraining process.

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

## A  THE FADING MEMORY EFFECT

Recent studies of hallucinations in VLMs have observed a "fading memory effect", where the model's sensitivity to the visual input declines as text generation proceeds, and its outputs gradually rely more heavily on learned language priors rather than on the image. As the attention to visual tokens decays over time, the model tends to drift—introducing non-existent objects, attributes, or relations not grounded in the image Xie et al. (2025); Favero et al. (2024). Although these phenomena were originally characterized in earlier VLMs (e.g. the first generation of LLaVA Liu et al. (2023)), they persist in more advanced models as well. In Figure 10, we show the same diagnostic computation from fig. 2, but applied to Qwen2.5-VL Qwen (2025); Wang et al. (2024) — one of today's most powerful open-source VLMs — thereby demonstrating that the fading memory effect still manifests in state-of-the-art systems.

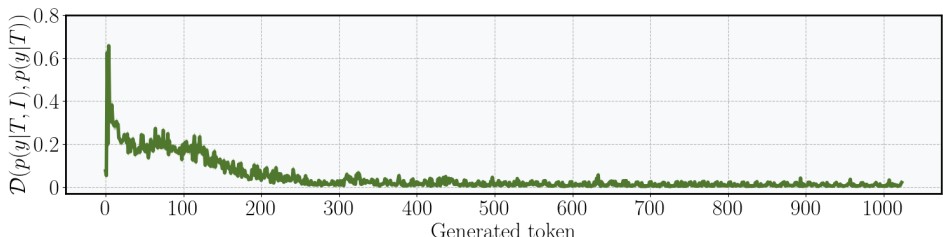

Figure 10: The "fading memory effect" in Qwen2.5-VL Qwen (2025). We calculate the next token prediction with and without the visual input, and we compute the distributional difference. After the first tokens, the next token can be predicted just from the previously generated text.

## B  ReCo vs OTHERS

ReCo is uniquely placed in the intersection of methods that treat the underlying VLM as a black box (e.g., Leng et al. (2024); Woo et al. (2024)) and methods that are training-driven (e.g., Zhao et al. (2023); Liu et al. (2024c)). In table 4, we show the qualitative advantage of ReCo over multiple other proposed solutions for hallucination mitigation.

Table 4: ReCo compared to other models.

| Model | Training based | Black-box VLM | Can be deployed during VLM training | Single inference pass |
|---|---|---|---|---|
| OPERAHuang et al. (2024) | ✗ | ✗ | ✗ | ✗ |
| VCDLeng et al. (2024) | ✗ | ✓ | ✗ | ✗ |
| AvisCWoo et al. (2024) | ✗ | ✓ | ✗ | ✗ |
| M3IDFavero et al. (2024) | ✗ | ✓ | ✗ | ✗ |
| HALCJiang et al. (2024) | ✗ | ✓ | ✗ | ✗ |
| VTILiu et al. (2024c) | ✓ | ✗ | ✗ | ✓ |
| HA-DPOZhao et al. (2023) | ✓ | ✗ | ✓ | ✓ |
| **ReCo** | ✓ | ✓ | ✓ | ✓ |

## C  TRAINING DATA

We train ReCo using HA-DPO Zhao et al. (2023). The dataset consists of quadruples $(I, P, C, R)$ where $I$ corresponds to the image and $P$ corresponds to its accompanying prompt (or question). Finally, $C$ and $R$ correspond to *Chosen* and *Rejected* respectively: these are the two answers used for contrastive loss (or Direct Preference Optimization Rafailov et al. (2023)). An example can be seen in fig. 11. The whole dataset consists of only 1853 images which were extracted from the Visual Genome database Krishna et al. (2017).

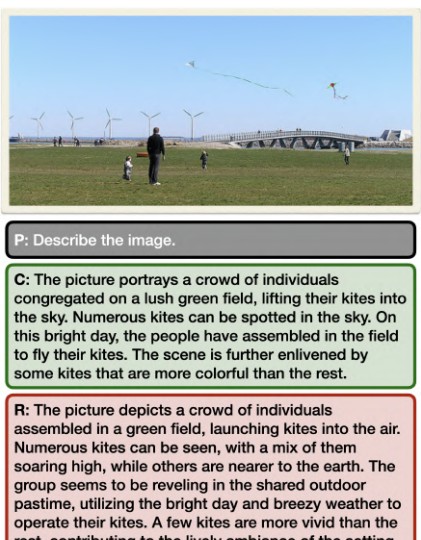

Figure 11: A sample of the HA-DPO dataset Zhao et al. (2023). While the *C* and *R* prompts may not be entirely accurate since they were generated with the help of LLMs Zhao et al. (2023), the Direct Preference Optimization Rafailov et al. (2023) successfully trains ReCo.

## D  HYPERPARAMETERS

We combine ReCo with two widely used VLMs: InstructBLIP Dai et al. (2023) and MiniGPT4 Zhu et al. (2023) and we follow the same training procedure for both. The learning rate is set to $5e$-3 and we train the model for 10 epochs. Additionally, the $\beta$ and $\lambda$ parameters of HA-DPO are set to $0.8, 0.2$ respectively. Finally, we chose a batch size of 128.

## E  BENCHMARKS

We consider three benchmarks designed to evaluate the hallucinating performance of VLMs across different tasks.

1. **POPE** Li et al. (2023): The POPE benchmark is focused on binary existence questions. It consists of 9000 questions in total, of the form *"Is there a ⟨object⟩ in the image?"*, each one accompanied by an image. POPE is designed to measure the disriminative capabilities of the underlying VLM, in the context of existence yes/no questions. The objects that inform the questions are chosen from the whole universe of the depicted objects in three different ways:

   - *Random*: randomly chosen from any of the existing objects in the dataset.
   - *Popular*: the objects are chosen from the top-k most frequent objects.
   - *Adversarial*: the objects are chosen based on the co-occuring frequencies with the objects depicted on each image.

   As expected, the "difficulty" of the benchmark increases as we transition from *Random* to *Adversarial*, something reflected in our results as well as the existing works too.

2. **CHAIR** Rohrbach et al. (2018): While POPE is focused on the discriminative capabilities of the VLMs, CHAIR assesses their generative power. This benchmark consists of images accompanied by their ground truth labels, i.e., which objects exist in each one of them. The VLM is prompted with a prompt such as *"Describe the image."*, or *"Provide a detailed description to the image."*. In our experiments we used *"Describe the image."* but any such prompt can be used with no changes. After each generated description is obtained, CHAIR estimates the hallucination rate of the VLMs, reporting two metrics:

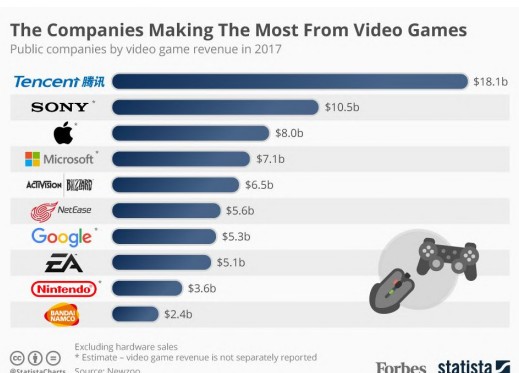

Figure 12: An image obtain from HallusionBench Guan et al. (2024). The corresponding question is: "According to the image, in 2017, was Tencent the company with the highest revenue from video games, with Sony as the second-highest earner?"

- *CHAIR$_i$*: CHAIR$_i$ is define as the ratio of the objects in the description that do not really exist in the image, i.e.,

$$\text{CHAIR}_i = \frac{|\{\text{hallucinated objects}\}|}{|\{\text{all mentioned objects}\}|} \tag{7}$$

- *CHAIR$_s$*: On the contrary, CHAIR$_s$ estimates how much the model "talks" about hallucinated objects, and it is defined as:

$$\text{CHAIR}_s = \frac{|\{\text{sentences with hallucinated objects}\}|}{|\{\text{all sentences}\}|} \tag{8}$$

3. **AMBER** Wang et al. (2023): AMBER combines and extends CHAIR and POPE, by evaluating the underlying VLM on both a wide array of discriminative questions about objects relationships (e.g., *"Is there a direct contect between $\langle object_1 \rangle$ and $\langle object_2 \rangle$"*), objects attributes (e.g., *"Is the cloud black in this image?"*), and object existence (*"Is there a $\langle object \rangle$ in the image?"*), as well as generative questions (*"Describe the image."*). The AMBER metric is calculated as:

$$\text{AMBER} = \frac{1}{2}\left(100 - \text{CHAIR} + \text{F1}\right) \tag{9}$$

with CHAIR$_i$ employed for the generative questions and F1 for the discriminative questions.

4. **MME** Fu et al. (2023): MME is yet another, larger, discriminative benchmark. Similarly to AMBER, MME also consists of multiple types of discriminative questions such as *artwork* related, *code* related, and *numerical* related questions. For each image, MME proposes two questions (one whose correct answer is *'Yes'* and one *'No'*). The total score for each type of questions is the summation of accuracy and accuracy+, with accuracy+ being the ratio of images in which both corresponding questions were answered correctly. Finally, the total perception score is the summation of all the corresponding scores for each question type.

5. **HallusionBench** Guan et al. (2024): Similar to MME, HallusionBench introduces a wide array of different discrimination questions. Differently than the other benchmarks, most of the questions in this benchmark are about charts, tables, and maps depicted on the given images. An example of a image-question pair can be seen in fig. 12. In many cases, the depicted diagrams do not accurately reflect reality and they were twisted in order to evaluate whether a VLM is actually "looking" at the image or its answer is based solely on the backbone LLM. Although the questions are discriminative, the long, detailed questions may lead to a more severe *fading memory effect*, compared to other discriminative benchmarks like POPE.

Table 5: POPE Li et al. (2023) results on MSCoco Lin et al. (2014) and A-OKVQA Schwenk et al. (2022), for Llava Liu et al. (2023). In both datasets, ReCo provides a significant performance boost to all the methods (where *Original* stands for the unmodified Llava).

| | Model | MSCoco Lin et al. (2014) | | | | | | A-OKVQA Schwenk et al. (2022) | | | | | |
| | | Random | | Popular | | Adversarial | | Random | | Popular | | Adversarial | |
| | | Acc (↑) | F1 (↑) | Acc (↑) | F1 (↑) | Acc (↑) | F1 (↑) | Acc (↑) | F1 (↑) | Acc (↑) | F1 (↑) | Acc (↑) | F1 (↑) |
|---|---|---|---|---|---|---|---|---|---|---|---|---|---|
| Llava Liu et al. (2023) | Original | 84.7% | 85.0% | 81.8% | 82.6% | 76.7% | 78.7% | 81.9% | 83.5% | 75.9% | 79.2% | 68.4% | 74.5% |
| | **+ReCo** | +0.5% | -0.3% | +1.3% | +0.2% | +2.5% | +1.1% | +2.4% | +1.3% | +3.0% | +1.5% | +2.6% | +0.9% |
| | VCD | 85.1% | 85.4% | 81.6% | 82.6% | 76.2% | 78.6% | 81.8% | 83.6% | 75.5% | 79.1% | 67.8% | 74.3% |
| | **+ReCo** | +2.4% | +1.5% | +2.3% | +1.2% | +3.3% | +1.6% | +3.5% | +2.3% | +5.1% | +3.2% | +4.3% | +2.0% |
| | M3ID | 86.4% | 86.5% | 82.8% | 83.5% | 77.3% | 79.3% | 83.0% | 84.6% | 76.7% | 79.9% | 68.6% | 74.7% |
| | **+ReCo** | +0.7% | -0.3% | +2.4% | +1.1% | +3.5% | +1.4% | +4.5% | +3.1% | +6.0% | +3.8% | +6.0% | +2.9% |
| | AvisC | 87.8% | 87.8% | 83.9% | 84.5% | 78.2% | 80.1% | 84.5% | 85.8% | 78.6% | 81.4% | 69.0% | 75.2% |
| | **+ReCo** | +0.8% | +0.2% | +2.9% | +2.0% | +3.5% | +2.0% | +3.4% | +2.4% | +2.9% | +1.6% | +4.0% | +1.9% |

Table 6: POPE Li et al. (2023) results on GQA Hudson & Manning (2019), for all three VLMs (InstructBLIP Dai et al. (2023), MiniGPT4 Zhu et al. (2023), and Llava Liu et al. (2023)).

| | Model | GQA Hudson & Manning (2019) | | | | | |
| | | Random | | Popular | | Adversarial | |
| | | Acc (↑) | F1 (↑) | Acc (↑) | F1 (↑) | Acc (↑) | F1 (↑) |
|---|---|---|---|---|---|---|---|
| InstructBLIP Dai et al. (2023) | Original | 79.2% | 80.7% | 73.2% | 76.6% | 68.8% | 73.5% |
| | **+ReCo** | +3.4% | +0.1% | +4.7% | +0.1% | +7.0% | +1.5% |
| | VCD Leng et al. (2024) | 80.9% | 81.8% | 73.3% | 76.3% | 69.6% | 74.1% |
| | **+ReCo** | +2.9% | +0.1% | +6.3% | +2.1% | +6.2% | +1.0% |
| | M3ID Favero et al. (2024) | 81.0% | 82.2% | 74.4% | 77.4% | 69.3% | 74.0% |
| | **+ReCo** | +3.4% | +0.1% | +5.1% | +0.6% | +7.1% | +1.3% |
| | AvisC Woo et al. (2024) | 85.1% | 85.3% | 76.2% | 78.5% | 72.0% | 76.0% |
| | **+ReCo** | +0.4% | -0.3% | +4.0% | +0.8% | +5.1% | +0.9% |
| MiniGPT4 Zhu et al. (2023) | Original | 53.9% | 32.9% | 52.6% | 32.3% | 51.7% | 31.5% |
| | **+ReCo** | +3.0% | +30.0% | +2.7% | +18.7% | +2.9% | +19.8% |
| | VCD Leng et al. (2024) | 55.7% | 38.8% | 54.1% | 38.0% | 53.4% | 37.7% |
| | **+ReCo** | +0.9% | +11.8% | +0.9% | +11.8% | +0.1% | +11.7% |
| | M3ID Favero et al. (2024) | 55.3% | 37.9% | 54.0% | 37.2% | 52.5% | 35.7% |
| | **+ReCo** | +1.5% | +15.5% | +1.4% | +15.4% | +1.6% | +15.8% |
| | AvisC Woo et al. (2024) | 63.1% | 63.5% | 60.0% | 61.6% | 57.0% | 60.0% |
| | **+ReCo** | -2.4% | -13.5% | +0.7% | -12.0% | +0.2% | -14.1% |
| LLaVA Liu et al. (2023) | Original | 82.3% | 84.0% | 73.9% | 78.1% | 68.6% | 74.8% |
| | **+ReCo** | +2.5% | +1.6% | +2.3% | +0.9% | +4.2% | +1.9% |
| | VCD Leng et al. (2024) | 81.9% | 84.0% | 72.5% | 77.5% | 68.1% | 74.9% |
| | **+ReCo** | +4.7% | +3.2% | +3.7% | +1.8% | +4.5% | +1.9% |
| | M3ID Favero et al. (2024) | 83.6% | 85.3% | 74.3% | 78.7% | 68.8% | 75.1% |
| | **+ReCo** | +3.8% | +2.3% | +5.5% | +2.8% | +6.6% | +3.3% |
| | AvisC Woo et al. (2024) | 84.6% | 86.1% | 74.6% | 79.0% | 69.4% | 75.8% |
| | **+ReCo** | +3.4% | +2.3% | +5.1% | +3.0% | +6.0% | +3.2% |

# F  MORE QUANTITATIVE RESULTS

## F.1  POPE

Besides table 2 in which we present the POPE results for InstructBLIP Dai et al. (2023) and MiniGPT4 Zhu et al. (2023) on MSCoco and A-OKVQA Lin et al. (2014); Schwenk et al. (2022), in table 5 you can observe the results on the same datasets for Llava Liu et al. (2023) too. The observations from the main text hold here too, with ReCo being a valuable addition to all the considered baselines, improving the performance by as much as 6% (absolute improvement). Finally, in table 6 you can observe the results for all three VLMs in yet another dataset: GQA Hudson & Manning (2019). Similarly to the previous observations, ReCo consistently improves the results across the board in this dataset too.

## F.2 CHAIR

In table 7, we depict the improvement that ReCo offers over all baselines in the generative task of image description, when the underlying VLM is LlaVA Liu et al. (2023). Similar to the observations of the main text, ReCo reduces both CHAIR$_s$ and CHAIR$_i$, especially as the output length increases, with the improvement being as high as 44% and 33%, respectively.

Table 7: CHAIR Rohrbach et al. (2018) results for Llava Liu et al. (2023). *Original* stands for the unmodified VLM, while VCD Leng et al. (2024), M3ID Favero et al. (2024), and AvisC Woo et al. (2024) are the three baselines we consider. In all four cases, the addition of ReCo improves significantly the performance as the generation progresses, reducing CHAIR$_s$ as much as 44% and CHAIR$_i$ 33%, with 32, 64, 128, 256, 512, and 1024 standing for the maximum allowed number of generated tokens.

| Model | | CHAIR$_s$(↓) | | | | | | CHAIR$_i$(↓) | | | | | | Average length |
|---|---|---|---|---|---|---|---|---|---|---|---|---|---|---|
| | | 32 | 64 | 128 | 256 | 512 | 1024 | 32 | 64 | 128 | 256 | 512 | 1024 | |
| Llava Liu et al. (2023) | Original | 7.8 | 23.8 | 50.0 | 50.6 | 50.6 | 50.6 | 4.5 | 8.2 | 15.3 | 15.3 | 15.3 | 15.3 | 500 |
| | **+ReCo** | +4.6 | +0.4 | -1.4 | -5.4 | -16.8 | -22.6 | +2.8 | +1.3 | -0.3 | -0.2 | -3.9 | -5.1 | 319 |
| | VCD | 8.0 | 22.6 | 52.2 | 48.4 | 48.4 | 48.5 | 4.4 | 7.7 | 16.0 | 15.3 | 15.3 | 15.3 | 486 |
| | **+ReCo** | -0.4 | +2.4 | -10.4 | -8.0 | -15.9 | -13.6 | -0.7 | +1.1 | -3.1 | -2.9 | -3.3 | -4.0 | 306 |
| | M3ID | 8.2 | 22.4 | 55.4 | 57.2 | 57.2 | 57.2 | 3.9 | 6.9 | 15.8 | 16.2 | 16.2 | 16.2 | 495 |
| | **+ReCo** | +1.2 | -0.2 | -6.2 | -16.8 | -19.0 | -24.8 | +0.7 | +0.7 | -2.6 | -4.2 | -5.2 | -5.7 | 335 |
| | AvisC | 9.2 | 21.0 | 53.6 | 55.8 | 55.8 | 55.8 | 5.3 | 6.9 | 15.3 | 17.1 | 17.1 | 17.1 | 523 |
| | **+ReCo** | -0.1 | +2.0 | -0.2 | -8.8 | -9.2 | -9.8 | -0.3 | +1.1 | +1.3 | -2.7 | -3.6 | -3.5 | 428 |

## F.3 MME

Besides POPE, which is restricted to existence-related questions, we evaluate ReCo in the context of multiple types of discriminative questions, using the MME benchmark. Although, just like POPE, such a benchmark does not directly assess the effect (*fading memory effect*) we are trying to eradicate in this work, it is important to examine whether the addition of ReCo (or any other component) hurts the model's performance in such discriminative tasks. In table 8, we can observe that ReCo not only preserves but rather improves the performance of InstructBLIP Dai et al. (2023) and MiniGPT4 Zhu et al. (2023), with or without the addition of any of the other baselines. Regarding Llava Liu et al. (2023), the addition of ReCo preserves the already amazing performance, while at the same time, as we showed in table 7, it reduces dramatically its hallucinating performance on the free text generation.

Table 8: MME Wang et al. (2023) results for InstructBLIP Dai et al. (2023), MiniGPT4 Zhu et al. (2023), and Llava Liu et al. (2023). ReCo consistently improves the performance of InstructBLIP and MiniGPT4, while the performance of Llava remains at the same level.

| Model (+ReCo) | MME (↑) | | |
|---|---|---|---|
| | InstructBLIP | MiniGPT4 | LLaVA |
| Original | 1355 (+150) | 939 (+051) | 1543 (-019) |
| VCD Leng et al. (2024) | 1495 (+062) | 933 (+115) | 1582 (+045) |
| M3ID Favero et al. (2024) | 1458 (+080) | 961 (+061) | 1619 (+031) |
| AvisC Woo et al. (2024) | 1373 (+121) | 879 (-023) | 1661 (-051) |

## G MORE QUALITATIVE RESULTS

In the following figures, we present more examples for both generative and discriminative questions. In many of the cases, the VLMs fail to provide a cohesive answer. On the contrary, ReCo is able to significantly improve the results, as it is apparent by the reduction of hallucinating objects, unrelated-to-image text, as well as by the accurate answering of the discriminative questions.

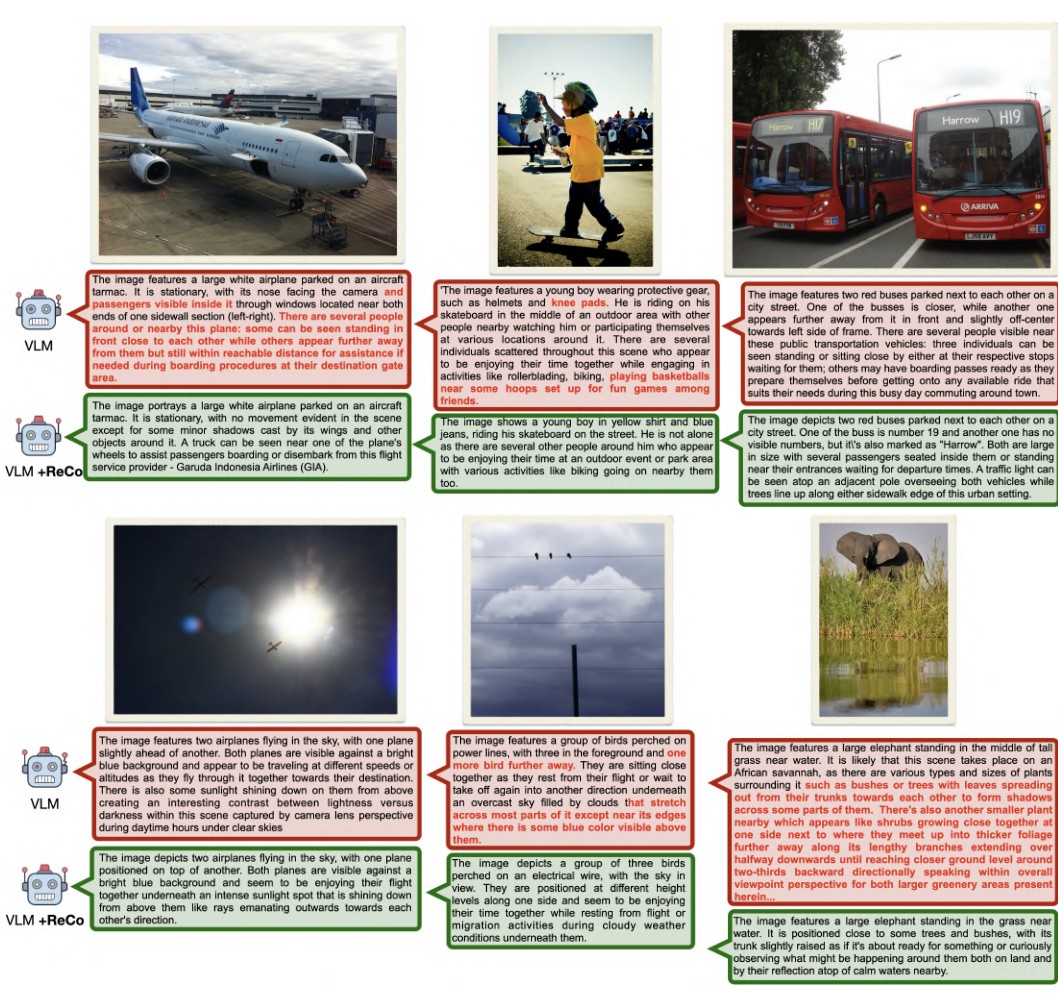

Figure 13: Generative questions on InstructBLIP Dai et al. (2023). The prompt used is *"Describe the image."*

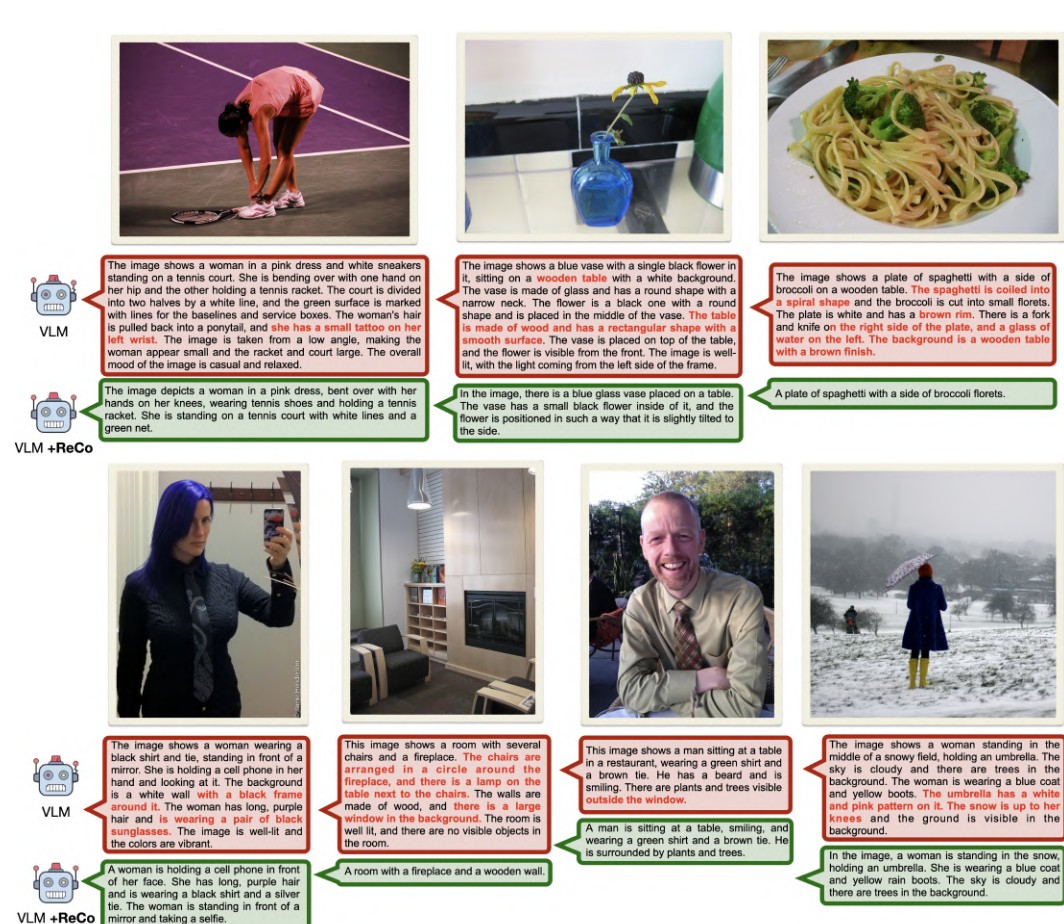

Figure 14: Generative questions on MiniGPT4 Zhu et al. (2023). The prompt used is *"Describe the image."*

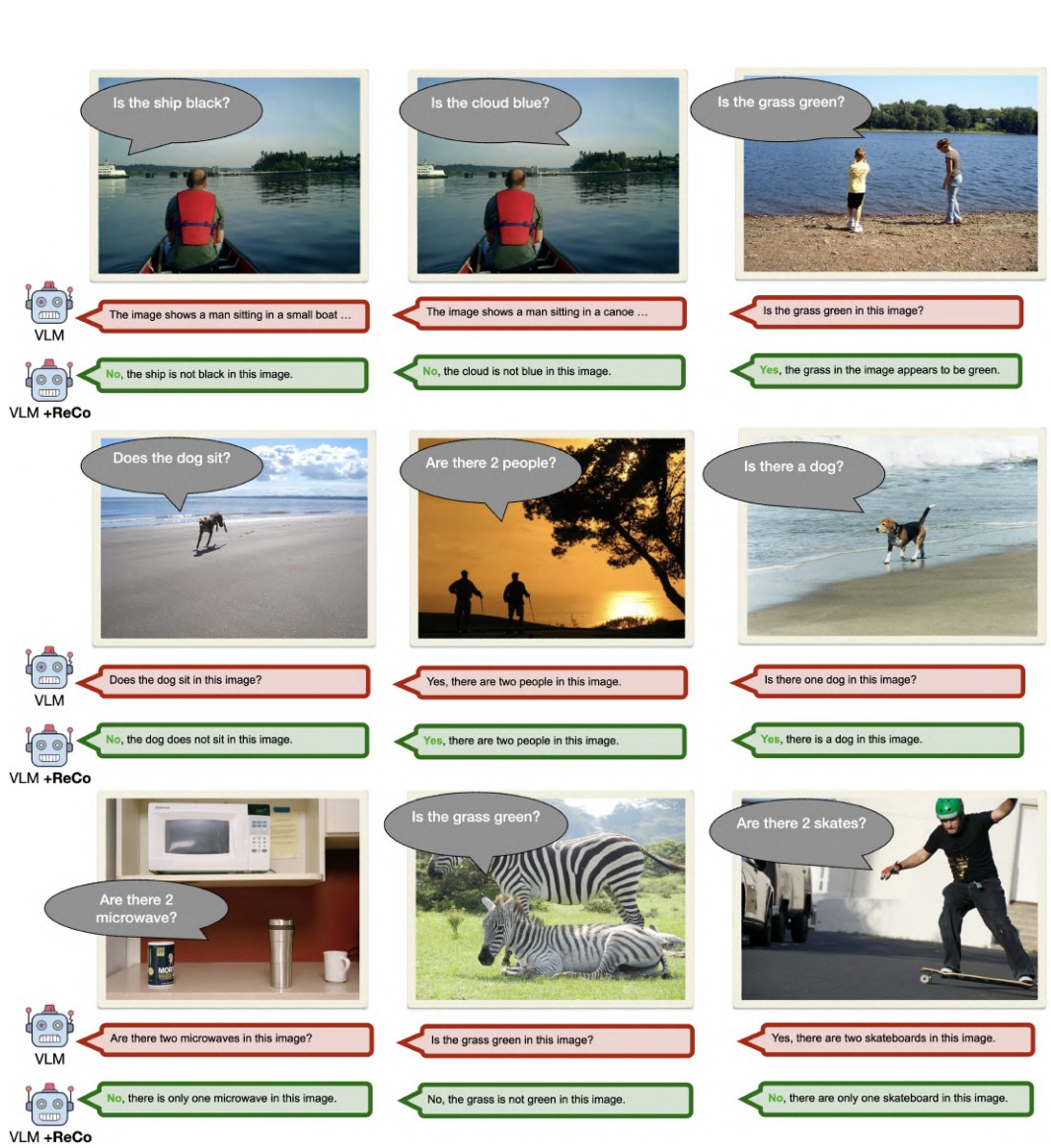

Figure 15: Discriminative questions on MiniGPT4 Zhu et al. (2023), taken from the AMBER benchmarks. The unmodified VLM fails to answer cohesively in most of the cases, while the addition of ReCo (although not trained on such data) allows the model to answer correctly.

