# OpenReview forum: "ReCo: Reminder Composition Mitigates Hallucinations in Vision-Language Models"
_ICLR.cc/2026/Conference — ICLR 2026 Conference Withdrawn Submission_

### Official Review · Reviewer_tQ4u · 2025-10-18

**Soundness:** 3
**Presentation:** 2
**Contribution:** 2
**Rating:** 4
**Confidence:** 4

**Summary:**

The paper introduces ReCo, a small trainable module that plugs into existing Vision-Language Models (VLMs) to counter their tendency to hallucinate—outputs that aren’t grounded in the image. Motivated by the observed fading memory effect (visual information’s influence diminishes as text generation proceeds), ReCo draws on ideas from geometric algebra and relational composition to explicitly recombine visual and textual signals before each next-token prediction, without modifying the base VLM. Applied to InstructBLIP, LLaVA, and MiniGPT-4, ReCo yields consistent gains across multiple benchmarks. The module is complementary to other hallucination-mitigation techniques, further improving results when combined.

**Strengths:**

- ReCo adds a tiny trainable layer before the prediction head, requiring no changes to the base VLM, minimal training, and negligible deployment overhead.


- Demonstrates robust improvements across multiple VLMs (InstructBLIP, LLaVA, MiniGPT-4) and benchmarks (e.g., CHAIR, POPE, AMBER, HallusionBench), indicating broad applicability.


- Stacks cleanly with prior decoding/mitigation techniques and yields further gains, supporting practical integration into real systems.

**Weaknesses:**

- The images are too —they’re not vector graphics.
- The citation format also looks incorrect.
- Ablations don’t isolate the “composition” effect，such as gains from W_T​ vs. W_I​, image-token pooling choices, and alternative operators aren’t disentangled.
- Most of the benchmarks used in the paper are discriminative. Consider adding generative benchmarks as well—for example, FaithScore: Fine-grained Evaluations of Hallucinations in Large Vision-Language Models.

- Motivation is unclear: the paper does not convincingly justify why geometric-algebra/VSA–style set/composition operations are needed here.
- In addition, performance should be compared against training-based methods, such as Aligning Large Multimodal Models with Factually Augmented RLHF, FGAIF: Aligning Large Vision-Language Models with Fine-Grained AI Feedback, and Mitigating Object Hallucination in Large Vision-Language Models with Human-Free Reinforcement Learning. The paper also omits some training-free baselines, for example Woodpecker: Hallucination Correction for Multimodal Large Language Models and A Unified Hallucination Mitigation Framework for Large Vision-Language Models.

**Questions:**

see weakness

---

### Official Review · Reviewer_pxDz · 2025-10-20

**Soundness:** 2
**Presentation:** 2
**Contribution:** 2
**Rating:** 2
**Confidence:** 4

**Summary:**

This paper proposes Reminder Composition (ReCo), a lightweight module designed to mitigate the fading memory effect in Vision-Language Models (VLMs). ReCo operates right before the language head, re-injecting visual embeddings at every decoding step to reinforce visual grounding. The process is formulated using Geometric Algebra (GA) and implemented as a small trainable layer, which can be trained independently or jointly with the entire model. Experimental results show consistent performance improvements across all evaluated VLMs, and ReCo can be combined with other hallucination-mitigation approaches such as VCD, M3ID, and AvisC. Furthermore, qualitative analyses demonstrate that ReCo not only reduces object-related hallucinations but also alleviates structural hallucinations.

**Strengths:**

- The paper provides an interpretation of the fading memory effect in VLMs through the lens of Geometric Algebra, offering a clear and conceptually motivated formulation of the proposed ReCo module.
- The proposed method demonstrates consistent improvements across evaluated VLMs, supported by both quantitative and qualitative experiments that validate its effectiveness in reducing hallucinations and enhancing visual grounding.

**Weaknesses:**

- The proposed method appears conceptually similar to Li et al. (ICML 2025), “The Hidden Life of Tokens: Reducing Hallucination of Large Vision-Language Models via Visual Information Steering.” A more explicit discussion of the similarities and differences between ReCo and this prior work would clarify the paper’s unique contribution.
- Continuously re-injecting image features at every decoding step may risk making the visual signal overly dominant, potentially reducing the language model’s contextual or semantic flexibility. An analysis of this trade-off would strengthen the argument.
- The transition from the Geometric Algebra formulation (Eq. 5) to the implementation equation (Eq. 6) feels abrupt. A more detailed explanation of how the theoretical formulation maps to the practical design, and especially what constitutes the “small trainable layer,” would improve clarity.
- Line 229: When the model is fine-tuned with $B_t$, it is unclear whether a distribution mismatch might occur at inference when only $T_t$ (without the reminder composition) is provided.
- To demonstrate the general applicability of the proposed module, it would be valuable to evaluate it on a wider range of VLMs, including more recent and stronger models such as Qwen-VL or InternVL. This would clarify whether ReCo’s gains hold universally or are primarily effective on lower-performing baselines.
- Including a user study for quantitative evaluation would further substantiate the qualitative claims and help assess perceived improvements in hallucination reduction.
- The paper lacks an ablation study that isolates the contribution of ReCo’s design. Without such analysis, it is unclear whether the improvement stems from the proposed structure itself or simply from an attention refresh effect. It would be informative to explore how performance varies with respect to the injection timing/location of visual features and the weighting parameters $W_T$ and $W_I$.
- In Section 4.2.2, the combination with AvisC seems less effective compared to other methods, yet the paper does not discuss why this interaction may be weaker.
- In Section 4.2.5, including a text-only baseline would make the interpretation of results clearer and help isolate the visual contribution.
- (Minor) References are currently written inline as plain text rather than enclosed in parentheses, which slightly disrupts readability. Consistent formatting, e.g., POPE (Li et al., 2023), would improve presentation quality.

**Questions:**

Please refer to Weakness.

---

### Official Review · Reviewer_sYHB · 2025-10-26

**Soundness:** 2
**Presentation:** 1
**Contribution:** 3
**Rating:** 4
**Confidence:** 3

**Summary:**

This paper proposes ReCo, a lightweight and plug-and-play module designed to mitigate the fading memory effect in Vision-Language Models (VLMs). By explicitly enforcing compositional reasoning between visual and textual representations, ReCo effectively reduces hallucination behaviors without modifying the backbone architectures. The paper provides extensive experiments across multiple benchmarks and shows consistent improvements over several baselines. Overall, this work is well-motivated and technically sound, but certain aspects of presentation, experimental depth, and comparison to recent methods could be further improved.

**Strengths:**

1. The paper provides a clear and intuitive explanation of the fading memory effect, supported by a well-designed visualization in Figure 2. This helps readers quickly grasp the core problem that ReCo aims to solve.
2. The introduction and theoretical background sections are detailed and logically structured, giving readers a solid understanding of the motivation behind Reminder Composition.
3. Experiments are conducted across five diverse benchmarks, demonstrating the general effectiveness of ReCo in mitigating hallucinations across multiple VLM architectures (InstructBLIP, LLaVA, MiniGPT-4).
4. The proposed approach is model-agnostic and lightweight, requiring minimal additional training, which makes it easy to integrate into existing systems.

**Weaknesses:**

1. In Figure 2, it is unclear which model the attention maps are derived from, and what the corresponding input data and generated tokens are. Clarifying these details would help readers better understand the relationship between visual attention and generated content.
2. While the introduction is well-written, it feels somewhat verbose. The authors might consider streamlining it and improving the logical transitions between paragraphs—for example, the sudden shift to the “Compositionality and Geometric Algebra” section disrupts the flow.
3. The comparison section mainly includes three hallucination mitigation baselines from roughly a year ago. It would strengthen the work to include more recent methods such as {Mitigating Hallucination of Large Vision-Language Models via Dynamic Logits Calibration} and {Reducing Hallucinations in Large Vision-Language Models via Latent Space Steering}.
4. The experimental analysis would benefit from more detailed ablation studies and diagnostics. For example, analyzing sensitivity to hyperparameters, conducting error analysis, and comparing visual attention strength before and after applying ReCo could provide deeper insights into its behavior.
5. While ReCo’s cross-model applicability is partially validated on InstructBLIP, LLaVA, and MiniGPT-4, it would be valuable to evaluate the method on more recent and advanced models such as Qwen2.5-VL or InternVL2.5 to assess scalability.
6. Minor issue: All figures in the manuscript appear blurry. Using vector graphics could greatly improve readability, especially for small text and fine visual details.

**Questions:**

Please see weaknesses

---

### Official Review · Reviewer_Dm4D · 2025-10-31

**Soundness:** 2
**Presentation:** 2
**Contribution:** 2
**Rating:** 2
**Confidence:** 4

**Summary:**

The goal of the paper is to mitigate hallucination in VLMs. The paper begins by describing the fading memory effect (also known as language prior or textual inertia), wherein VLMs progressively lose attention to visual tokens during the generation process. To balance visual and textual information, the paper considers the compositionality. Starting from Eq. (4), the paper introduces ReCo, which is composed of a linear combination of visual tokens and the current text tokens, thereby ensuring cross-modal attention during generation. The paper applies ReCo to the last layer using just two additional linear layers, which is minimal overhead in terms of parameters and computation. The paper shows the effectiveness of ReCo on hallucination benchmarks.

**Strengths:**

**S1. Efficient.** ReCo requires a low number of parameters and is located on the final hidden representation, resulting in low computational overhead.

**S2. Easy to Implement.** The method is straightforward to integrate, requiring only two lines of code.

**S3. Effective Across Diverse Hallucination Benchmarks.** The proposed approach demonstrates the effectiveness across multiple benchmark datasets.

**Weaknesses:**

**W1. Ambiguity of the black box.** Generally, we cannot access the last hidden layer in black-box models, such as GPT and Claude. With these black-box models, we can obtain only the generated results, namely the text. Thus, the proposed method cannot be used for black box models.

**W2. Comparison with training methods.** Hallucination mitigation methods can be categorized into training-based and training-free approaches. The proposed method falls within the training-based approach. The paper requires a fair comparison with other training-based methods (M3ID + DPO, HACL). Showing performance gains when applied to contrastive decoding is insufficient; a direct comparison with existing training-based methods is needed.

**W3. General task capabilities.** The proposed method requires training. Does ReCo preserve the general capabilities of VLMs across diverse general tasks?

**W4. Lack of Ablation Study.** In the context of relational composition, the attention is matrix binding operator. Thus, cross-attention is also possible to balance compositionality.

**Questions:**

While the paper provides an overview of Geometric Algebra, its relevance and contribution to the subsequent analysis are not entirely clear.

---

### Note · Authors · 2025-11-23

I have read and agree with the venue's withdrawal policy on behalf of myself and my co-authors.